# Exploring Metaphorical Transformations of a Safety Boundary Wall in Virtual Reality

**DOI:** 10.3390/s24103187

**Published:** 2024-05-17

**Authors:** Haozhao Qin, Yechang Qin, Jianchun Su, Yang Tian

**Affiliations:** Guangxi Key Laboratory of Multimedia Communications and Network Technology, School of Computer, Electronics and Information, Guangxi University, Nanning 530004, China; haozhaoqin@st.gxu.edu.cn (H.Q.); yechangq@st.gxu.edu.cn (Y.Q.); 2113391057@st.gxu.edu.cn (J.S.)

**Keywords:** virtual reality, interruptions, metaphor, safety boundary wall, awareness

## Abstract

Current virtual reality (VR) devices enable users to visually immerse themselves in the virtual world, contributing to their limited awareness of bystanders’ presence. To prevent collisions when bystanders intrude into VR users’ activity area, it is necessary to intuitively alert VR users to the intrusion event and the intruder’s position, especially in cases where bystanders intrude from the side or behind the VR user. Existing intruder awareness cues fail to intuitively present the intrusion event in such cases. We propose a novel intruder awareness cue called “BrokenWall” by applying a metaphor of “a wall breached by invading soldiers” to the VR user’s safety boundary wall. Specifically, BrokenWall refers to a safety boundary wall with a gap appearing in front of a VR user and rotating, guiding the user’s attention toward an intruder coming from the side or behind the VR user. We conducted an empirical study (N = 30) comparing BrokenWall with existing awareness cue techniques, Halo and Radar. Halo employs a sphere to represent the intruder, with the size indicating proximity and the position reflecting the direction. Radar employs a radar map to visualize the intruder’s position. The results showed that the BrokenWall awareness cue not only significantly reduces the time needed for users to detect an intruder but also has superior performance in subjective ratings. Based on our findings, we have established a design space for an interactive safety boundary wall to facilitate interactions between VR users and bystanders.

## 1. Introduction

When immersed in a virtual environment, VR users have limited awareness of bystanders due to the visual disconnect from the real world. When bystanders come near VR users, it may lead to accidental collisions [1,2], such as the VR users striking bystanders with their controller [3]. Therefore, when bystanders approach, VR users hope to receive immediate notifications about the presence and location of bystanders [2], especially in cases where bystanders approach from the side or behind the VR user. Do et al. [4] introduced Halo and Radar to indicate the presence and location information of bystanders in such cases. Halo employs a sphere to represent bystanders, with the sphere’s size indicating proximity and its position (left/right) reflecting the bystander’s direction relative to the user. Radar utilizes a radar map to visualize a bystander’s position. However, it is difficult for Halo and Radar to intuitively present the occurrence of intrusion events, thereby increasing the cognitive load on users.

To ensure the safety of VR users during their activities, popular head-mounted displays (HMDs) such as the Oculus Quest require users to delineate the boundaries of an empty physical activity area, i.e., the safety boundary [5,6]. The visualization of the safety boundary typically takes a virtual wall, referred to as a safety boundary wall, which serves to separate the virtual and real worlds. When users approach the safety boundary wall, its appearance serves as a warning of potential collisions with physical obstacles or bystanders outside the activity area. Users have the option to either halt their VR activities and return to the real world or retreat back into the delineated activity area to continue the VR experience. Inspired by the shape and warnings of the safety boundary wall, we propose a novel intruder awareness cue called “BrokenWall” by applying a metaphor of “a wall breached by invading soldiers” to the users’ safety boundary wall; an illustration is shown in Figure 1. Specifically, BrokenWall refers to when a bystander intrudes upon the activity area from the side or behind the VR user, and a safety boundary wall with a gap in front of the VR user appears. The appearance of the safety boundary wall represents the occurrence of an invasion in an intuitive manner. Then, the safety boundary wall rotates, shifting the gap towards the intrusion location to guide the VR user to notice the intruder.

In this work, we conducted an empirical user study to compare the BrokenWall cue with two other awareness cues (Halo and Radar) and evaluated these three intruder awareness cues. The results of the user study showed that regardless of the type of intruder (people/pets), with the BrokenWall cue, the participants required significantly less time to detect intruders than with the other two awareness cues. Through user surveys and interviews, we drew two conclusions: (i) BrokenWall reduces users’ cognitive load through the transformations of the safety boundary wall based on a metaphor of “a wall breached by invading soldiers”, and (ii) transformations of the safety boundary wall can serve as awareness cues to enhance interactions between VR users and bystanders. Consequently, we discussed and established a design space of awareness cues based on the safety boundary wall to facilitate VR users’ interactions with the bystanders.

## 2. Related Work

In this section, we discuss the following three research areas related to our work.

### 2.1. The Importance of Bystander Awareness

Dao et al. [3] conducted a classification study on online “VR fail videos”, revealing some of the reasons for which accidents may occur while engaging in VR activities. One of the reasons is that VR users have inadequate perception of the real world in virtual environments, leading to collisions with bystanders, physical walls, and furniture. Their research emphasizes the need to enhance the real-world awareness of VR users to prevent such collision events. Subsequently, O’Hagan et al. [1] conducted a survey on in-the-wild interactions between VR users and bystanders. The survey categorized the interactions between VR users and bystanders into three types: coexisting, demoing, and interrupting. Their survey outlined common impediments that may arise during these three types of interactions, such as the user being tripped by pets while immersed in VR activities. Their conclusions emphasize the significant role that bystanders play in interactions with VR users due to the limited real-world awareness of VR users. Building upon prior research, O’Hagan et al. [2] investigated the attitudes and expectations of VR users towards systems that facilitate interaction with the real environment. Their research findings reveal a range of user expectations and attitudes regarding the design of awareness-enhancing systems. Consequently, it is recommended that VR system designs incorporate features to strengthen users’ perception of the real world. For instance, systems should promptly notify VR users when people or pets are nearby, accurately indicating their locations.

### 2.2. Bystander Awareness Cues in Virtual Reality

The visual disconnect from the real world caused by VR devices has increased the likelihood of accidents involving VR users and their surroundings. To address this issue, McGill et al. [7] designed a system that integrates real-world objects and people into the virtual environment, allowing VR users to interact with them. Inspired by McGill et al.’s work, numerous studies have explored methods of informing VR users about the presence of bystanders through text notifications [8,9,10,11,12], audio notifications [8,9,11], haptic feedback [9], various avatar views [4,13,14,15,16], various forms of see-through videos [7,8,14,16,17], etc. However, for bystanders or pets that approach from the side or from behind VR users, VR users cannot immediately and accurately observe their position, even when virtual avatars or see-through videos are displayed in VR [13]. This contradicts the desire of VR users to be promptly informed of the positions of bystanders or pets [2,8]. Kudo et al. [13] compared the effectiveness of three different types of bystander awareness cues (Radar map, avatar view, and Presence++) in VR content with varying interactivity. Their findings showed that the Radar map provided better information on bystander location, while the virtual avatar representation enhanced VR users’ bystander awareness. Additionally, the size of the symbolic icon in Presence++ indicates proximity, while an icon with eyes reflects that a bystander is facing the VR user. Presence++ was more effective in maintaining user immersion. Do et al. [4] addressed privacy concerns between VR users and bystanders by designing a system that enables VR users to observe bystanders using three awareness cues: Halo, Radar, and Passthrough. Passthrough allows users to see the outside environment through the VR device’s camera. VR users wear an LED light band with three colors corresponding to the three awareness cues. Bystanders can identify the awareness cue being used by the VR user by observing the color of the light band, thus avoiding being recorded by the VR device’s camera without their knowledge.

### 2.3. Metaphors in VR Visual Cues

HCI research is primarily centered on the communication mediated between users and systems, where the use of metaphorical information representations serves to enhance understanding of a system [18]. Metaphors, by establishing associations with the real world, can effectively enhance users’ comprehension, learning efficiency, and interactivity with virtual environments [19]. Englmeier et al. [20] proposed the concept of a Spherical World in Miniature (SWIM) based on the tiny planet metaphor. By mapping virtual scenes onto the surface of a physical sphere, teleportation in VR can be achieved through rotation, translation, and stationary operations on the sphere. Compared with a planar WIM with a traditional VR controller technique, SWIM demonstrated superior performance in task completion time, accuracy, and subjective ratings. Wagner et al. [21] investigated the adaptation and evolution of traditional data visualization within immersive 3D environments based on VR technology. For a large dataset of taxi trips, they introduced the Space–Time Cube metaphor to describe data distributions in temporal and spatial dimensions. They believe that appropriate 3D interaction metaphors can help users better utilize 3D views by offering a more intuitive way to explore complex datasets. Pointecker et al. [22] presented five initial concepts for notification metaphors to bridge the gap between virtual environments and the real world. One of the concepts that they showcased involves placing a virtual door within the virtual environment. Upon opening this door, VR users can visually perceive the real world beyond the virtual door. They concluded that visual design can significantly impact the perception of metaphorical notifications. Ghosh et al. [9] effectively combined haptic, audio, and visual modalities in their design notifications for five VR interruption scenarios. By leveraging real-world metaphors, they introduced the concepts of footsteps and watches to intuitively represent the presence of bystanders and convey time, respectively. Informed by user surveys, design explorations, and experimental findings, they provided design recommendations for VR notifications, including the use of reality-based metaphors to enhance the comprehensibility of VR notifications when appropriate. Numerous metaphorical designs for interacting with real environments in VR have been developed; for example, Microsoft introduced the concept of a flashlight metaphor that allows users to peek into the real world through a virtual flashlight [23]. George et al. [24] introduced the metaphor of a virtual phone, enabling users to capture photos of their real-life surroundings. Medeiros et al. [16] employed the metaphors of shadows and ghost avatars to intuitively represent the locations of bystanders, among other examples.

### 2.4. Safety Boundaries in Virtual Reality

While users are immersed in VR experiences, a safety boundary wall serves to protect them in the physical world. Both the Oculus Guardian system [5] and HTC Vive’s Chaperone system [6] employ translucent walls to alert VR users when approaching safety boundaries, thus preventing them from venturing beyond activity areas. Cirio et al. introduced Magic Barrier Tape [25] based on the concept of hybrid position/rate control [26]. Magic Barrier Tape informs the user about the boundaries of their walking space. When users approach the edge of their walking space, they can rapidly move by “pushing” the virtual Barrier Tape. Qu et al. [27] introduced a dynamic safety boundary wall capable of predicting VR users’ motion amplitude in real time, enabling dynamic adjustment of the activity area to ensure user safety. Yang et al. [28] introduced a SharedSpace system that focuses on providing safety boundary walls for physical obstacles and bystanders rather than VR users. The SharedSpace system allows external observers to use physical “shield tools” to place safety boundary walls in the virtual world, expressing their needs for physical space to VR users. To address the challenge of insufficient physical space for real walking in VR, VirtualSpace [29] investigated the concept of dynamic safety boundaries for multiple VR users within the same physical environment. Their method dynamically allocates physical activity space for each VR user, promoting more efficient space utilization. Wu et al. [30] explored four safety boundary awareness techniques on smartphones or smartwatches, with the goal of preventing bystanders from invading the VR user’s activity area. These techniques involve displaying augmented reality boundary overlays and using visual, auditory, and haptic alerts to show how far the bystander is from the VR user’s activity area. The results demonstrated the efficacy of all four techniques, with augmented reality boundary overlays yielding the shortest walking time and haptic alerts proving to be the least disruptive for users. Prior research has typically utilized safety boundary walls to remind VR users to avoid straying outside of safe boundaries or to caution bystanders against intruding into VR users’ activity areas. Based on the metaphor of “a wall breached by invading soldiers”, we symbolize the occurrence of intrusion events through the appearance of a safety boundary wall with a gap in front of the VR user. Our approach augments the safety boundary wall’s functionality, extending its capabilities beyond providing alerts for VR users’ actions to include offering awareness cues about bystanders, thereby improving VR users’ sense of security and their perception of the real world.

## 3. User Study

In this user study, we examined the effects of two factors on user performance: the awareness cue factor and the intruder factor. The awareness cue factor had three levels (BrokenWall, Halo, and Radar), while the intruder factor had two levels (people and pets). Do et al. [4] introduced Halo and Radar to indicate the presence and location information of the bystanders. We selected Halo and Radar as candidate awareness cues, and their design is described in Section 1. Halo refers to a green sphere that appears on the (left/right) side of the VR user when an intruder enters the activity area from the side or behind the VR user, signaling the occurrence of an intrusion event. The sphere’s size shows how close the intruder is, and its relative position (left/right) to the VR user represents the intruder’s orientation relative to the VR user; see Figure 2a–c. Radar refers to how when an intruder enters the activity area from the side or from behind the VR user, a radar map appears directly in the upper middle of the VR user’s field of view. In the radar map, a red dot represents the VR user, and a blue dot represents the precise position of the intruder relative to the user; see Figure 2d–f.

***Participants.*** We recruited 30 participants (26 males and 4 females) from our campus, aged between 18 and 26 years (average age of 21.5 years). Among them, 22 participants had prior VR experience, and each participant received a compensation of 30 RMB.

***Apparatus.*** The experiments were conducted in a closed laboratory with an area of 80 square meters (8 m × 10 m). Within the laboratory, there was a 36 square meter (6 m × 6 m) open space, and the participants were positioned at the center for the experiment. We utilized the Vive Index VR headset as our experimental hardware and developed our experimental software using Unity 2019.4 (LST).

***Task.*** The task for each participant was to quickly detect the intruder during a VR game by using the provided awareness cues. The procedure was as follows. First, the participant needed to walk into the guidance circle at the center of the activity area. Next, the participant played a block-throwing game where they picked up blocks from a virtual table using a controller and tossed them onto a blue target block labeled “+1” to score points, aiming to accumulate as many as possible. At a random moment during the game, a virtual avatar of the intruder approached from the side or behind the participant. At this point, the participant needed to locate the intruder based on the received awareness cue. Upon spotting the intruder, the participant was required to immediately press the “B” button on the controller, and then the task ended; an illustration is shown in Figure 3.

**Experimental design.** We compared the BrokenWall awareness cue with the Halo and Radar awareness cues. We designed a block-throwing game that requires high visual attention and interaction with objects in a virtual environment for our experiment. We defined a circular safety boundary wall with a radius of 2 m in the virtual world. For BrokenWall, we set the height of the safety boundary wall to be consistent with each participant’s head height, ensuring uniform perception of the safety boundary wall by participants. At the center of the safety boundary wall, there was a 20 cm radius guidance circle where each participant was required to stand. Near the guiding circle, there was a virtual table with a fixed virtual marker placed in front of it to ensure that participants always looked in the same direction in the virtual environment, following the method used in prior work [7,8,14]. The distance and vertical speed of the target block were randomized to make the game more unpredictable, challenging, and engaging for the participants. In order to avoid predictability of intrusion events by participants, we adopted a random number N, ranging from 10 to 15. When the participant threw the Nth block, a virtual intruder avatar moved toward them at a steady speed of 1 m/s. The avatar started 3 m away and followed a straight path, getting closer to the participant until it was only 1 m away.

Before the experiment started, we provided detailed explanations to the participants about the experiment’s purpose, informing them that they would engage in a virtual reality game and experience three different awareness cues to enhance their ability to detect intruders. Each participant performed the task as described previously under six conditions (three levels of the awareness cue factor (Halo, Radar, and BrokenWall) and two levels of the intruder factor (people and pets)). The orders of the levels of the two factors were counterbalanced across participants using balanced Latin square designs. As illustrated in Figure 4, we divided the safe activity area into two semicircles. In the semicircle outside the participants’ field of view, we defined five intruder routes, each separated by a 45° angle and corresponding to a separate trial. Each participant completed 6 sessions of trials, covering all possible combinations of the two factors. Each session consisted of two practice trials to help the participants become familiar with the task and five blocks of the formal trial. Participants took a three-minute rest after each session. The entire experiment lasted approximately one hour. Consequently, our experimental design (excluding practice trails) had a total of 30 participants × 6 blocks × 5 trials = 900 trials.

The parametric dependent variable was the average time, which was the mean value of all time periods from the appearance of the awareness cue for the participants, ending when they pressed the button on the controller in a session. After each session, we asked participants to provide ratings on the NASA-TLX form and to answer a questionnaire. The NASA-TLX form was used to assess the workload. The questionnaire included the following seven statements on a seven-point Likert scale (1: strongly disagree, 7: strongly agree): (i) I think this awareness cue feels natural within the VR environment (Naturalness); (ii) I think it is easy to understand this awareness cue (Understandability); (iii) I think this awareness cue conveys a sense of urgency (Urgency); (iv) I think this cue enhances the sense of security in the VR experience (Security); (v) I think this awareness cue is effective (Efficiency); (vi) I think this awareness cue makes me feel comfortable (Comfort); (vii) I like this awareness cue (Like). In addition, the experimenter engaged in one-on-one discussions with each participant and recorded their comments on their experience during the user study.

**Results.** Figure 5 and Table 1 show the results. After conducting a normality test, it was found that the average time data did not follow a normal distribution. Therefore, the Aligned Rank Transformation (ART) test with post hoc Wilcoxon signed rank tests was used to analyze the average time data. For all subjective evaluation data, the non-parametric Friedman test with post hoc Wilcoxon signed rank tests was used for analysis.

***Average Time.*** There was a significant interaction between awareness cue factors and intruder factors $\left(F_{2,58}=5.942,\;p<0.01\right)$. Significant differences were found between the awareness cue factor levels $\left(F_{2,58}=77.603,\;p<0.001\right)$, and there were also significant differences between the intruder factor levels $\left(F_{1,29}=7.395,\;p<0.05\right)$. The average time for BrokenWall (M = 1.97 s, SD = 0.6) was significantly shorter than for Halo (M = 2.37, SD = 0.6, Z = −4.10, p<0.001) and Radar (M = 2.66 s, SD = 0.7, Z = −4.78, p<0.001) when the intruder was a pet. The average time for Halo was significantly shorter than for Radar (Z = −3.05, p<0.01) when the intruder was a pet. The average time for BrokenWall (M = 1.88 s, SD = 0.5) was significantly shorter than for Halo (M = 2.13, SD = 0.5, Z = −3.80, p<0.001) and Radar (M = 2.63 s, SD = 0.8, Z = −4.78, p<0.001) when the intruder was a human. The average time for Halo was significantly shorter than for Radar (Z = −4.56, p<0.001 ) when the intruder was a human.

***Naturalness.*** We found significant differences ($\chi^{2}\left(2\right)=21.0,\;p<0.001$). The Naturalness score for BrokenWall (M = 5.9, SD = 1.6) was significantly higher than those for Halo (M = 4.9, SD = 1.1, Z = −2.87, p<0.01) and Radar (M = 4.3, SD = 1.0, Z = −3.85, p<0.001). The Naturalness score for Halo was significantly higher than that for Radar (Z = −2.35, p<0.05).

***Understandability.*** We found significant differences ($\chi^{2}\left(2\right)$ = 15.45, p<0.001). The Understandability score for BrokenWall (M = 6.1, SD = 1.0) was significantly higher than those for Halo (M = 5.1, SD = 1.7, Z = −2.70, p<0.01) and Radar (M = 4.6, SD = 1.3, Z = −3.51, p<0.001).

***Urgency.*** We found significant differences ($\chi^{2}\left(2\right)=38.26,\;p<0.001$). The Urgency score for BrokenWall (M = 6.3, SD = 0.8) was significantly higher than those for Halo (M = 5.2, SD = 1.2, Z = −3.18, p<0.01) and Radar (M = 4.0, SD = 1.1, Z = −4.66, p<0.001). The Urgency score for Halo was significantly higher than that for Radar (Z = −3.58, p<0.001).

***Security.*** We found significant differences ($\chi^{2}\left(2\right)=33.81,\;p<0.001$). The Security score for BrokenWall (M = 6.2, SD = 0.8) was significantly higher than those for Halo (M = 4.3, SD = 1.5, Z = −4.14, p<0.001) and Radar (M = 4.1, SD = 1.0, Z = −4.70, p<0.001).

***Efficiency.*** We found significant differences ($\chi^{2}\left(2\right)=21.87,\;p<0.001$). The Efficiency score for BrokenWall (M = 5.9, SD = 1.1) was significantly higher than those for Halo (M = 5.1, SD = 1.0, Z = −2.02, p<0.05) and Radar (M = 4.1, SD = 1.1, Z = −4.05, p<0.001). The efficiency score for Halo was significantly higher than that for Radar (Z = −3.17, p<0.01).

***Comfort.*** We found significant differences ($\chi^{2}\left(2\right)=19.83,\;p<0.001$). The Comfort score for BrokenWall (M = 5.9, SD = 1.2) was significantly higher than those for Halo (M = 4.6, SD = 1.4, Z = −3.32, p<0.01) and Radar (M = 4.3, SD = 1.5, Z = −3.26, p<0.01).

***Like.*** We found significant differences ($\chi^{2}\left(2\right)=28.75,\;p<0.001$). The Like score for BrokenWall (M = 6.1, SD = 1.0) was significantly higher than those for Halo (M = 4.9, SD = 1.3, Z = −2.92, p<0.01) and Radar (M = 4.2, SD = 1.1, Z = −4.39, p<0.001). The Like score for Halo was significantly higher than that for Radar (Z = −2.20, p<0.05).

***Workload.*** We found significant differences ($\chi^{2}\left(2\right)=40.492,\;p<0.001$). The Workload score for BrokenWall (M = 5.2, SD = 2.6) was significantly lower than those for Halo (M = 6.8, SD = 2.3, Z = −4.15, p<0.01) and Radar (M = 9.0, SD = 2.5, Z = −4.62, p<0.001). The Workload score for Halo was significantly lower than that for Radar (Z = −4.60, p<0.05).

***Qualitative Feedback.*** The majority of the participants (26 out of 30) expressed a preference for the BrokenWall awareness cue. They commented that BrokenWall offered a more easily understandable and intuitive awareness cue. The metaphor of a breached safety boundary wall quickly informed them of intrusion events. On the other hand, since the safety boundary wall is a familiar setting, BrokenWall had a lower learning cost than that of the other two awareness cues. One participant’s comment (P7) reflected the views of most participants, stating, “When immersed in the game scenario, I found that BrokenWall’s warning effect allowed me to quickly understand the occurrence of intrusion events compared to the other two awareness cues. This helped me to promptly stop my actions and respond to the awareness cue accordingly. In contrast, Halo and Radar did not convey the same sense of urgency to me. Regarding locating the intruder’s position, Radar required more time to understand and think about the relative positions of myself and the intruder to determine their direction. For Halo, I needed to turn in the direction where it appeared. For BrokenWall, I could easily follow the rotation of the safety boundary wall, which guided me to locate the intruder effortlessly”. When asked which awareness cue made them feel more comfortable, 28 participants chose the BrokenWall awareness cue. A representative comment from one participant (P20) was, “I’m quite familiar with safety boundary walls, and the BrokenWall awareness cue gives me a psychological warning that my activity area has been intruded upon. Through the display of the safety boundary wall, I can also understand the positions of myself and the intruder in the real world, which makes me feel safer. This is especially evident when a pet enters, as the BrokenWall awareness cue allows me to detect pets more quickly and conveniently, reducing the chances of accidents caused by pet entry. I prefer awareness cues like the safety boundary wall fixed in one position, as opposed to cues like Halo and Radar that remain static in front of me relative to my perspective within the virtual environment. This is because the BrokenWall awareness cue feels less obstructive and more natural to me”.

***Summary.*** Our study found that the BrokenWall awareness cue significantly reduces the time for VR users to detect intruders compared to two existing awareness cues. Assigning metaphors to transformations in the safety boundary wall as an awareness cue effectively reduces the cognitive load on users.

## 4. Discussion

***BrokenWall reduces cognitive load***. When the intruder was a human, BrokenWall achieved an average detection time of 1.88 s, while Halo and Radar achieved average detection times of 2.13 s and 2.63 s, respectively. Similarly, when the intruder was a pet, BrokenWall achieved an average detection time of 1.97 s, compared to 2.37 s for Halo and 2.66 s for Radar. The reasons behind the results should be as follows. First, the metaphor of “a wall breached by invading soldiers” employed in BrokenWall enabled users to quickly and intuitively understand the occurrence of intrusion events. Halo and Radar, lacking a clear metaphor, might cause users to require additional time to interpret and process information from the awareness cues, potentially leading to delayed reactions or confusion. Second, users easily detected the intruder location with BrokenWall, as the rotation of the safety boundary wall shifted the gap and finally remained at the intrusion location, facilitating effortless identification for VR users. For Halo, the green sphere representing the intruder was fixed on either side of the user, causing the user to focus on turning to one side and potentially miss the pet avatar. For Radar, the constant shift in users’ attention between the radar map and the first-person-perspective environment resulted in a slow location of intruders.

***BrokenWall interrupts users’ VR activity more efficiently.*** The safety boundary wall surrounds VR users, easily drawing their attention with its expansive coverage area. Meanwhile, the appearance of the safety boundary wall carries a warning meaning. In our experiment, we observed that with the BrokenWall awareness cue, the participants always tended to stop their current activity (e.g., stop throwing blocks) immediately upon seeing the wall. With the other two awareness cues, the participants exhibited a different tendency, with more than half of them occasionally choosing to complete their current activity (e.g., throwing a block) before starting to locate the intruder.

***BrokenWall is adapted from an existing VR system element.*** Given that the safety boundary wall is a familiar VR system element for users, adapting it to be an awareness cue enables users to readily accept and easily distinguish it from the virtual elements in their current VR environment. On the other hand, transforming the safety boundary wall into an awareness cue does not introduce additional VR system elements. In contrast, other awareness cues, such as textual hint windows, Halo, Radar, etc., mean introducing new system elements, potentially increasing the system’s complexity.

## 5. Future Work: Exploring a Design Space for Interactive Safety Boundary Walls

Due to the advantages of the safety boundary wall in conveying information on real-world surroundings, we established a safety boundary wall design space to facilitate interactions between VR users and bystanders. The design space of the safety boundary wall includes two main dimensions.

Dimension 1: Interactions between VR users and bystanders. We categorize these interactions into three types: interruption, coexistence, and AR interaction. Interruption refers to instances where bystanders disrupt the user’s VR experience. Coexistence refers to scenarios where bystanders and VR users share the same physical space without interfering with each other’s activities. AR interaction refers to scenarios in which bystanders interact with VR users through the visualization of the VR users’ safety boundary walls using augmented reality technology.Dimension 2: Transformations of the safety boundary wall. We mainly classify transformations of the safety boundary wall into three types: rigid (e.g., translation, rotation, etc.), non-rigid (e.g., fragmentation, height changes, etc.), and texture changes (e.g., color, material, etc.).

Based on these two dimensions, we conducted an extensive brainstorming session to explore all potential possibilities within the safety boundary wall design space. In the end, we presented 18 examples in the form of a safety boundary wall design space, including 16 novel ideas (see Table 2). These examples provide valuable references for future research and practical applications. In the future, we plan to further investigate and assess the various possibilities within this safety boundary wall design space.

## 6. Conclusions

We propose the BrokenWall awareness cue, a novel virtual reality awareness cue based on a metaphor of “a wall breached by invading soldiers”, i.e., it displays a broken safety boundary wall to indicate an intrusion event and uses the rotation of the safety boundary wall to indicate the direction of the intruder. In this user study, we compared BrokenWall with the Halo and Radar awareness cues. The results showed that BrokenWall significantly reduces the time for VR users to detect intruders compared to the other two awareness cues. Assigning metaphors to transformations of the safety boundary wall as an awareness cue effectively reduces the cognitive load on users. Finally, combining our empirical results and participant feedback, we discussed and established a design space of awareness cues based on a safety boundary wall to enhance interactions between VR users and bystanders.

## Figures and Tables

**Figure 1 sensors-24-03187-f001:**
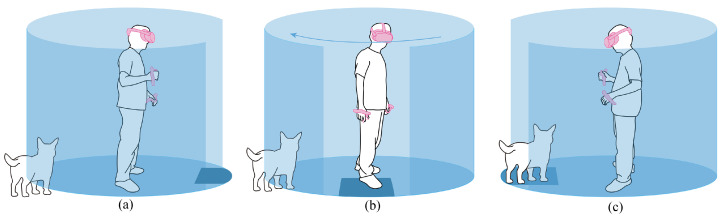
Illustration of the BrokenWall awareness cue. (**a**) A VR user is playing game. When a pet dog initially enters the activity area, a safety boundary wall with a gap in front of the VR user appears. (**b**) The VR user turns around as the safety boundary wall rotates. (**c**) The gap stops behind the dog, and a section of the safety boundary wall collapses beneath the dog’s feet, leading the VR user to notice the dog.

**Figure 2 sensors-24-03187-f002:**
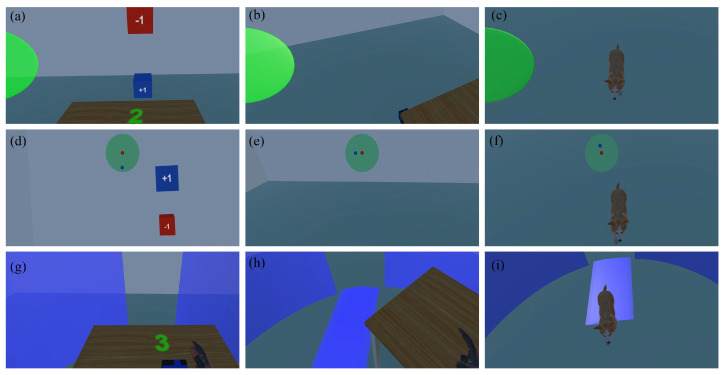
The three awareness cues. From top to bottom, the three rows represent the Halo, Radar, and BrokenWall awareness cues, respectively. (**a**) When the virtual avatar of the intruder enters the activity area from behind the VR user, a green sphere appears on the left. (**b**) The user turns to the left. (**c**) The user notices the intruder. (**d**) When the virtual avatar of the intruder enters the activity area from behind the VR user, a radar map appears above and in front of the user’s field of view. (**e**) The user turns to the left, and the relative positions of the two dots on the radar change accordingly. (**f**) The user notices the intruder. (**g**) When the virtual avatar of the intruder enters the activity area from behind the VR user, a broken safety boundary wall appears in front of the user’s field of view. (**h**) The user turns to the left as the safety boundary wall rotates. (**i**) The user notices the intruder.

**Figure 3 sensors-24-03187-f003:**
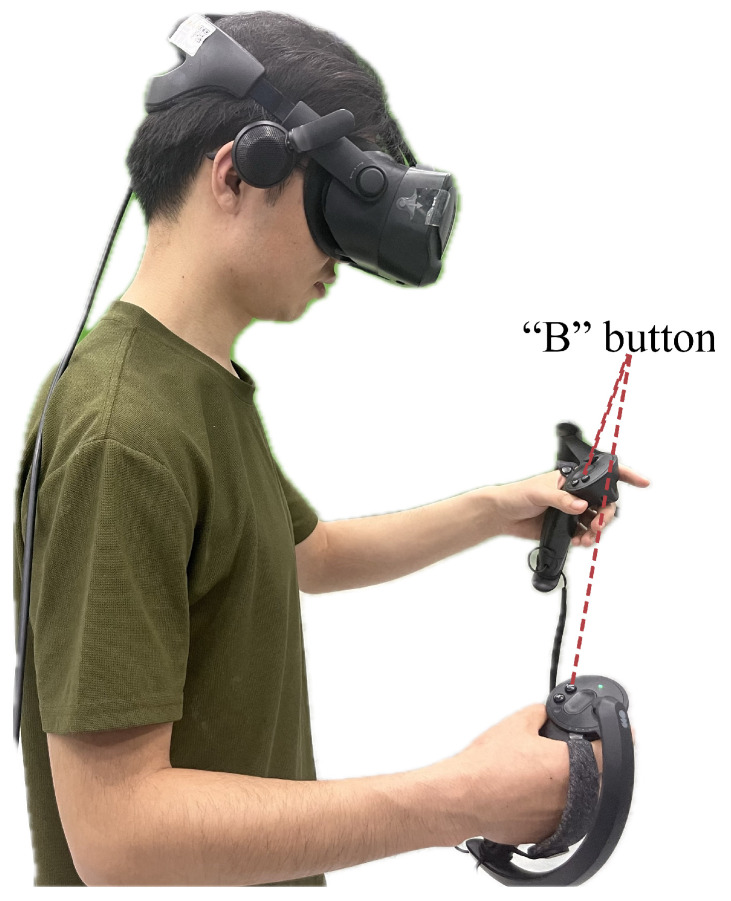
A participant is playing a block-throwing game in the virtual world.

**Figure 4 sensors-24-03187-f004:**
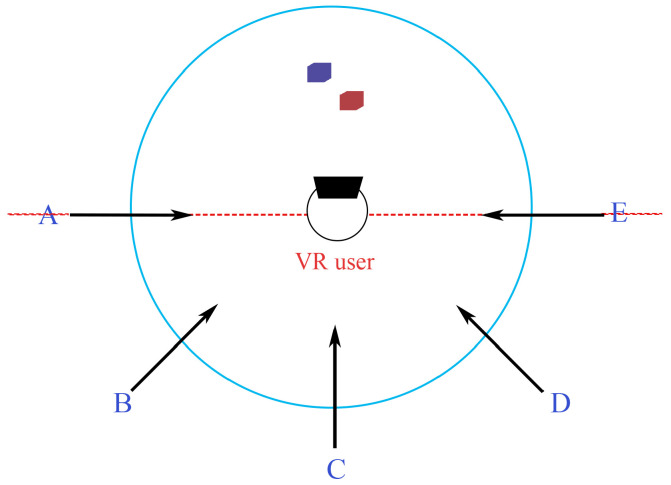
The five routes for intruders in the user study. A represents a route for the virtual avatar to enter the VR user’s activity area. B to E represent four more routes serving the same purpose.

**Figure 5 sensors-24-03187-f005:**
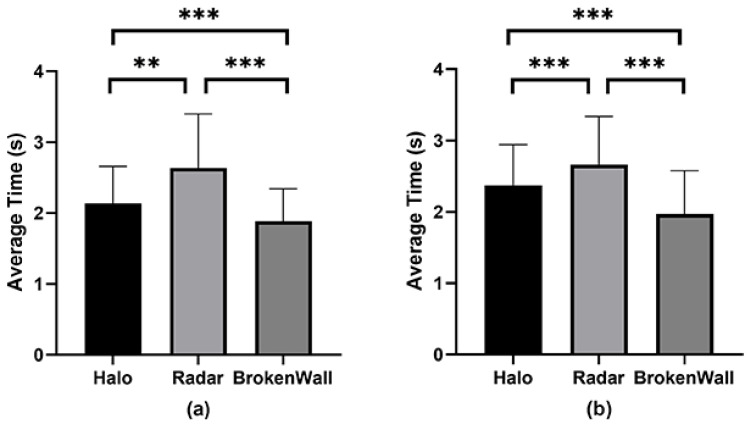
(**a**) The average time in the user study under different levels of the awareness cue factor when the intruder was a pet. (**b**) The average time in the user study under different levels of the awareness cue factor when the intruder was a human. ** means p<0.01. *** means p<0.001.

**Table 1 sensors-24-03187-t001:** The results of the user study.

Experimental Data	1 Halo	2 Radar	3 BrokenWall	Non-Parametric Friedman Test	*Post–hoc:*Wilcoxon
Naturalness	4.9 (1.1)	4.3 (1.0)	5.9 (1.2)	$\chi^{2}$(2) = 21.0 p<0.001	1-2, 1-3, 2-3
Understandability	5.1 (1.7)	4.6 (1.3)	6.1 (1.0)	$\chi^{2}$(2) = 15.453 p<0.001	1-3, 2-3
Urgency	5.2 (1.2)	4.0 (1.1)	6.3 (0.8)	$\chi^{2}$(2) = 38.264 p<0.001	1-2, 1-3, 2-3
Security	4.3 (1.5)	4.1 (1.0)	6.2 (0.8)	$\chi^{2}$(2) = 33.811 p<0.001	1-3, 2-3
Efficiency	5.1 (1.0)	4.1 (1.1)	5.9 (1.1)	$\chi^{2}$(2) = 21.868 p<0.001	1-2, 1-3, 2-3
Comfort	4.6 (1.4)	4.3 (1.5)	5.9 (1.2)	$\chi^{2}$(2) = 19.825 p<0.001	1-3, 2-3
Like	4.9 (1.3)	4.2 (1.1)	6.1 (1.0)	$\chi^{2}$(2) = 28.752 p<0.001	1-2, 1-3, 2-3
Workload	6.8 (2.3)	9.0 (2.5)	5.2 (2.6)	$\chi^{2}$(2) = 40.492 p<0.001	1-2, 1-3, 2-3

“# (#)” means the mean value and the corresponding standard deviation under a condition. “#-#” means that there was a significant difference between the numbered conditions.

**Table 2 sensors-24-03187-t002:** Summary of the safety boundary wall design space with two dimensions. The design space reveals 16 novel ideas and 2 examples from prior work. The examples from prior work are denoted with (*).

		Interactions between VR Users and Bystanders		
		Interruptions	Coexistence	AR Interaction
**Transformations of the safety boundary wall**	Rigid	Rotation: The safety boundary wall notifies the user about an intruder’s direction through rotation. Vibration: The safety boundary wall notifies the user about intrusion events through vibration.	Translation: The safety boundary wall adjusts its position to allow for some activity space for bystanders through translational motion. Vibration: The safety boundary wall adjusts its vibration intensity to display the magnitude of real-world noise.	Translation: Bystanders push the safety boundary walls of VR users to enable their movement during spatial conflicts.
	Non-rigid	Breaking: The breaking of the safety boundary wall notifies the user about intrusion events.	Height: Changes in the safety boundary wall’s height indicate the proximity of bystanders. Scale: During spatial conflicts, the safety boundary wall scales up or down to adjust the activity area [27]. (*) Deformation: During spatial conflicts, the safety boundary wall deforms to alter the activity area [25]. (*)	Deformation: Bystanders can deform the security boundary wall by squeezing it with their hands or bodies. Breaking: When bystanders touch the safety boundary wall with their hands or bodies, the safety boundary wall will break to remind them that they are entering the VR user’s activity area.
	Texture	Color: The safety boundary wall changes its color to signify intrusion events.	Blinking: Bystanders’ approach is indicated by the blinking of the safety boundary wall. Color: A darker color of the safety boundary wall indicates the proximity of bystanders. Transparent: The nearest section of the safety boundary wall is displayed to indicate the path of a bystander’s movement. Texture Mapping: Crack textures appear in security boundary walls to represent surrounding noise.	Color: The color of the safety boundary wall represents the current status of the VR user. Transparency: Changes in the transparency in the safety boundary wall indicate whether the VR user is demoing VR to bystanders. Texture Mapping: Changes in the texture mapping of the safety boundary wall indicate whether the VR user is demoing VR.

## Data Availability

Data are contained within the article.

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
