# Peer review of "Exploring Metaphorical Transformations of a Safety Boundary Wall in Virtual Reality"

_sensors, 2024, doi:10.3390/s24103187_

Round 1
Reviewer 1 Report
Comments and Suggestions for Authors
This empirical study showed that the BrokenWall awareness cue significantly reduces users' time in detecting the intruder. The content is well structured. There are some questions and suggestions as follows:
1. In the abstract, it is not necessary to explain the concept of BrokenWall by using a large text. Instead, the test process should be introduced briefly.
Section "2.1. The Importance of Bystander Awareness" is more like a study background. However, previous research on the linkage mechanism between vision and perception, as well as metaphors
has not been reviewed.
2. Will prior VR experience affect perceptual differences, as only 20 out of 30 participants have VR experience? Due to the limited number of participants, it is difficult to draw strong conclusions from this variable.
3. this empirical study's result can only prove that BrokenWall can provide a more robust perception, but why were some design suggestions that referred to color, material, etc., given? The discussion, as the most important part of the paper, is insufficient as the experimental results did not explain the previous research.
Comments on the Quality of English LanguageQuality of English Language is ok.
Reviewer 2 Report
Comments and Suggestions for Authors
Exploring Metaphorical Transformations of Safety Boundary Wall in Virtual Reality
This article proposes a novel concept for alerting users to intruders in virtual reality scenarios. The authors describe the current problem state: VR devices offer users visually immersive experiences in virtual environments, often leading to reduced awareness of nearby bystanders. To prevent collisions when bystanders enter a VR user's activity space, it's crucial to promptly alert users to the intrusion and the intruder's location, especially when intrusions occur from the sides or behind the VR user. Existing intruder awareness cues often fail to effectively convey intrusion events in these scenarios.
Thus, the authors introduce a new concept called "BrokenWall" for alerting users to potential intruders. This concept draws on the metaphor of "a wall breached by invading soldiers" to create a visual cue within the VR environment. Specifically, when a bystander enters the activity area from the side or behind the VR user, a gap appears in the safety boundary wall in front of the user. The wall then rotates, aligning the gap with the location of the intrusion, thus directing the user's attention towards the intruder. Results from their empirical study demonstrate that the BrokenWall cue significantly reduces the time required for users to detect intruders compared to existing cues.
Building on these findings, the authors have developed a design framework for interactive safety boundary walls to enhance interactions between VR users and bystanders. The 'Introduction' and 'Literature Reviews (Related Works)' sections are logically and clearly presented. Especially, Figure 1 Illustration of the BrokenWall (as called by the authors) awareness cue is visually and meaningfully presented, allowing readers to fully understand it. Furthermore, the concept of 'Safety boundary in Virtual Reality' is described in simple terms with visual Figure 2 showing the three awareness cues. An adequate 'User Study' was conducted to examine the effects of two factors on user performance: the awareness cue factor and the intruder factor. Overall, the methodology and procedures of experimental components are plainly presented. In the 'Results' section, appropriate statistical analysis has been presented with details (Figure 5 and Table 1). Additionally, the 'Discussion and Design Recommendations' and 'Conclusions' sections are clearly stated, specifically the summary of safety boundary wall design space described in detail in Table 2 (Interactions between VR user and bystanders).
Their METHODOLOGY that implemented was adequate. Additional usability activities will be helpful in future research.
Conclusions were defiantly consistent with the results and arguments presented and main research questions was clearly addressed.
Overall, this article appears to provide a novel and efficient technique for safety boundary walls in virtual reality.
Comments on the Quality of English LanguageMinor editing is needed...
Round 2
Reviewer 1 Report
Comments and Suggestions for Authors
I think this paper has been aligned with the publication standard.
Comments on the Quality of English LanguageQuality of English Language is ok.